# Challenging Ergonomics Risks with Smart Wearable Extension Sensors

**Nikola Maksimović [1,2], Milan Čabarkapa [3,4,\*], Marko Tanasković [1] and Dragan Randjelović [5]**

1   Faculty of Information Technology and Computing, Singidunum University, Danijelova 32, 11000 Belgrade, Serbia
2   Academy of Technical and Art Applied Studies, Department Textile School for Design, Technology and Management, 11000 Belgrade, Serbia
3   Faculty of Engineering, University of Kragujevac, 6 Sestre Janjić Street, 34000 Kragujevac, Serbia
4   Faculty of Electrical Engineering, University of Belgrade, 11000 Belgrade, Serbia
5   Faculty of Diplomacy and Security, University Union-Nikola Tesla Belgrade, 11000 Belgrade, Serbia
\*   Correspondence: cabmilan@etf.rs or mcabarkapa@kg.ac.rs

**Abstract:** Concerning occupational safety, the aim of ergonomics as a scientific discipline is to study and adjust working conditions, worker equipment, and work processes from a psychological, physiological, and anatomical aspect instead of adapting the worker to the needs of the job. This paper will discuss and analyze the potential of the garment-embedded body posture tracking sensor and its usage as standard working equipment, which is meant to help correct improper and high-risk upper body positions during prolonged and static work activities. The analysis evaluation cross-reference is based on the Rapid Upper Limb Assessment ergonomics risk assessment tool. Signals generated by the wearable are meant to help the wearer and observer promptly-continuously detect and correct bad posture. The results show a positive progression of workers' body posture to reduce the ergonomic risks this research covers. It can be concluded that wearable technology and sensors would significantly contribute to the observer as the evaluation tool and the wearer to spot the risk factors promptly and self-correct them independently. This feature would help workers learn and improve the correct habits of correcting ergonomically incorrect body postures when performing work tasks.

**Keywords:** ergonomics; sensors; wearables; occupational safety; posture analysis

## 1. Introduction

This paper aims to analyze the assumption that positive results are achievable by wearable technology as an aid in the prevention of ergonomic risk conditions and to evaluate the recorded results through one of the globally established methods for detecting ergonomic risks. Procedures and tools for studying and detecting ergonomic risks in the work environment have a challenge ahead of them that they should be minimally invasive in the sense that they do not interfere with the observed person and the activities he or she performs [1] at the same time they should provide realistic and representative usable data. The collected data is then used through various tried and tested methodologies for ergonomic risk assessment. Each of these methodologies aims to reduce emerging risk factors of work-related musculoskeletal disorders (WMSDs) and improve work in terms of ergonomics and productivity [1–5]. Ergonomic risk refers to physical stress factors and workplace conditions that carry the risk of damage or disease to the musculoskeletal systems of employees. In this paper, we use the Rapid Upper Limb Assessment RULA methodology [1,2], as our reference evaluation tool for upper body ergonomic risk assessment using the wearable sensor for posture risk evaluation. RULA is an ergonomics-based work risk assessment tool. Dr. Lynn McAtamney and Professor E. Nigel Corlett, ergonomists from the University of Nottingham in England, have developed it. In the ergonomic field of study, there are

other tools, standards, and metrologies, such as the U.S. National Institute for Occupational Safety and Health (NIOSH) lifting equation [1]; similar to the NIOSH tool, we have The Liberty Mutual MMH Tables ("Snook Tables") they cover various lifting, lowering, pushing, pulling, and carrying tasks based on research by Dr. Stover Snook and Dr. Vincent Ciriello at the Liberty Mutual Research Institute for Safety, "Snook Tables" is based on psychophysical assessment measures, not on biomechanical ones like with NIOSH. RULA is similar to the Rapid Entire Body Assessment Method (REBA) developed by Sue Hignett and Lynn McAtamney at Nottingham Hospital (The United Kingdom). Another method to mention here is the Hand Arm Risk Assessment Method (HARM), which has been developed for occupational health officers to perform risk assessments of developing arm, neck, or shoulder pain during hand arm tasks. The international standard for occupational safety is ISO 6385 [6], which establishes the fundamental ergonomics principles for work systems. A big challenge in applying these methods is collecting the correct data in practice. It relies on the experience and skill of an ergonomic expert to spot the risks and communicate the corrective advice back to the worker, or analyzes are performed in laboratory conditions that are not always relevant or applicable to the actual work conditions. The sensor we created is embedded in the back part of a regular work shirt, and its task is to collect data on spine flexion and trunk twisting. When the values are in the ergonomic risk zone, the microprocessor that monitors the sensor readings gives the wearer a sound and light signal to correct them. The thresholds of these values are calibrated with the recommended values of the RULA assessment tool [2].

The rest of this paper is organized as follows: the following Section 2. Literature Review will cover past studies we reviewed and used as references to our research. Then in Section 3. Materials and Methodologies: We give an overview of the procedures, measuring tools, and methods we used to get the results in Section 5. Results we present the comparative results of our research and their analysis. 6. Discussion is where we overview received results and discuss drawbacks and challenges. Section 6. Conclusions is where we give our final thoughts in this paper regarding occupational safety and wearables topic.

Contributions

The study's contribution is reflected in another potent research for the solutions that can help predict and prevent the ergonomic risks associated with WMSDs. The motivation was to multidisciplinary explore the integration of wearable sensors into regular garments, considering suitable materials for the final products that are anthropometrically fitted so they can be used as a tool to evaluate ergonomics risks conditions with REFA as a cross-reference. A different approach in designing sensors for detecting the risk thresholds is properly evaluating stretchable electroconductive paint on flexible textile surfaces to detect body movements. The emphasis was on minimal interference with the wearer so that the quality of collected data is not compromised by a worker feeling uncomfortable or examined. It raises the potential for everyday use in the real-world scenario that would inevitably reduce work-related diseases caused by bad ergonomic habits and environments. The sensor's microcontroller is of an open architecture kind. It can be facilitated with a wireless connection module and other modules or sensors, effectively transforming the whole product into a wearable IoT platform providing remote assessments and monitoring. At the same time, a new tool was tested that can help and facilitate the use of representative methods aimed at evaluating ergonomic risks when performing work tasks. In our case, it was RULA because we could get rapid results on site at the factory. However, a sensor like this can also provide valuable evaluation data for other methodologies. The sensor enables even less experienced ergonomists to perform better risk assessment tasks and workplace designs. The worker also receives immediate feedback, thus resulting in faster training and adopting good ergonomic habits, consequently contributing directly to occupational safety.

## 2. Literature Review

Occupational ergonomics promotes a holistic approach that considers physical, cognitive, social, organizational, environmental, and other relevant factors related to work-

related injuries' causes. As such, it requires solutions for its challenges in multidisciplinary sources [1–7]. We reviewed many studies in the past that have been evaluating the introduction of different types of sensors which can be an observatory, wearable, or a combination of both [7–12]. Wearable sensors can track environmental conditions [3,5,7], movements, or physiological data. The use of wearables in the ergonomics field aims to increase work efficiency among employees, improve their physical well-being, and reduce WMSDs [13–15]. From the ergonomics point of view, the wearable sensor must be such that it can produce the movement reading of the human body as anatomically correct and natural as possible [5,8,12]. An essential property of wearables is their flexibility to track and collect data anytime and anywhere [5,12]. It can be sad that wearable technology extends human capabilities bridging interactions of humans and technology [5,7], with the potential to help us overcome various physical and hazardous challenges [9,15]. This is very important considering modern industrial developments and evermore present human-machine interactions since we are entering the era of industry "4,0" [14], which is bringing yet another ergonomics risk due to increasingly static work tasks [15]. A better understanding of the role and the status of humans can facilitate and improve the overall human-machine system performance, as well as ensure the well-being of humans [14]. In human factors and ergonomics, conventional methods of human performance evaluation usually require the efforts of trained personnel for data collection, data analysis, and the explanation of results [1,15,16]. The development of wearable technologies has improved the potential for developing a more innovative and automatic solution to performing relevant evaluations of posture parameters [11,12]. Most of the current studies propose wearable solutions for assessing ergonomic risk factors. Deferent types of sensor systems, like insole pressure systems, were assessed [17,18] and also body-mounted smartphone solutions [19,20], are examined for producing exposure recognition or assessment of postures, activities, and excessive risks. Some [19,21] employ machine learning techniques and cloud architecture in the research. It is evident that the risk level analysis has attracted the attention of many studies [22–24], as well as possible answers to them [25–27]. The validation of solutions is mainly performed using established ergonomic methods [28,29]. Research in this field focuses on detecting and reducing factors that affect WMSDs globally [30]. From the point of view of employing deep learning techniques, studies are done to understand better behaviors [19], related to risk conditions. To understand ergonomic risk factors in general, research has been done to separate workers' risk population groups [31,32]. This research showed us that construction workers are some of the most affected by physically challenging tasks like this. Robotics is again considered to assist and correct the worker [33], in terms of exoskeleton propositions. Ergonomics dealing with challenges of illness prevention, and bad ergonomic practice leads to injuries or, more often, chronic conditions. Studies have been done to research rehabilitation devices [34,35], examining wearables as therapeutic tools.

## 3. Materials and Methodologies

The authors proposed their wearable sensor and microcontroller model as a material and method based on the Score Chart methodology. Data analysis was performed for two separate scenarios performed under industrial operating conditions. Upon the recorded data analysis was completed, a statistical analysis of the results was performed in order to verify the initial thesis of the study.

The research involved a series of recordings of manual work tasks in a manufacturing environment following ergonomics and work-study guidelines, with a worker wearing the garment carrying an embedded sensor while performing regular work tasks. Data collected while using the sensors were compared to data collected without wearable sensors for the same work tasks performed by the same person under the same work conditions using the RULA ergonomics risk assessment tool. For both cases in this comparison, the assessment data were taken in different periods during the worker's day shift and summarized in marked hours, taking into account the factor of worker fatigue and efficiency

at the timeframe level of the entire day shift lasting 7.5 h. The scale of this research is aimed to cover screening the trunk and neck posture separately from other risk conditions; the trunk and neck posture score is a part of the complete Rapid Upper Limb Assessment (RULA) tool. The values of force created by the worker during the evaluated task, which involved manipulation of tools and product items, were measured with a dynamometer and calculated together with the frequency of occurrence in the ergonomics risk assessment score calculator.

### 3.1. Materials

Our experiment used a stretch sensor for tracking body posture and a microcontroller that monitors the obtained data. The stretch sensor is a flexible capacitor that can give precise information about shape deformation. This is done by relating changes in capacitance to geometry according to the Parallel Plate equation:

$$C = \varepsilon_0 \, \varepsilon_r \, (A/d)$$

where C is the sensor's capacitance, A is the surface area, d is the thickness of the dielectric layer, $\varepsilon_0$ is the absolute permittivity, and $\varepsilon_r$ is the relative permittivity of the dielectric layer. The capacitance of a stretch sensor is directly proportional to the area of the parallel flexible electrodes and is inversely proportional to the distance between the flexible electrode layers. Stretching a sensor causes both the area and thickness to change. Variations in the strain of the sensor can be linearly converted to variations in capacitance, and then linearly converted to an output voltage; therefore, the strain can be calculated by the voltage. The microcontroller is based on the STM32L4 platform and is connected to the sensor via analog inputs where it reads its voltage. It is powered by a 3.3 V battery rsrv, the whole package is built into the garment in a special pocket in the seamline.

The sensor is based on electroconductive paint with stretch properties and is suitable for application on flexible textile fabrics and garments. The sensor, which acts as a conductor, is monitored by a microcontroller to detect changes in the electric resistance measured on the sensor. The microcontroller is programmed to detect any exceeded value of the thresholds and accordingly provide notification to the operator. Notifications are provided using sound and LED light signals depending on the current needs and conditions. Changes in the electric resistance are generally dependent on the conductor's cross-sectional area, the conductor, the length of the conductor, and its resistivity. We are using this property of the resistance to measure the changes in value readings on the microcontroller at the moment of beginning and during prolonged stretch periods.

### 3.2. Methods

Initial calibrations of the microcontroller reading are required to secure a provision of the valid data reading. Initial calibrations are aligned with the anthropometric parameters of the subject, which are then correlated with Rapid Upper Limb Assessment neutral thresholds values using presets or dynamometer readings, after this garment is ready to wear with the sensor actively detecting any deviation from the preset values at a programable rate of one reading per 0.1 s. The entire process is conducted in real-time in three phases. The first one is detecting the conductor value data, the second is processing and evaluating the collected data, and the last is adequate notification output. Garment size and sensor are matched to fit anthropometric body size measurements according to ISO 8559-1:2017 and ISO 8559-2:2017 approved by CEN (European Committee for Standardization). ISO 8559-2:2017 specifies primary and secondary dimensions for specified types of garments to be used in combination with ISO 8559-1 which covers anthropometric definitions for body measurement.

The preproduction garment pattern pieces are made-to-measure according to the data taken from ISO 8559-1:2017, the body measurements size charts for apparel industry standards. These body measurements, in our case, represent shoulders and neck width, chest circumference, waist circumference, hip circumference, and back length. Based on these body measurements, the sensor is placed on the back piece of the garment in such a

position that it can monitor the bending and twisting deformations of the wearer's back and spine.

The garment should be chosen and designed accordingly to the assessment tasks. We have chosen a regular work-style t-shirt for our research. Body size fitting and sensor positioning are coordinated with anthropometrical measurements of the target group of workers to precisely track spine flexion and twisting of the torso. Ease in the t-shirt fit is also a factor here. It must be tight enough for the sensor to read resistance oscillations correctly yet still adequately comfortable for the wearer. Knitted fabrics are chosen for the garment to correspond to the sensor stretch properties and not interfere with conductor resistance readings. Modifications in the garment design aim to fit the microcontroller, wiring, and battery pack comfortably and to protect sensitive parts of the wearable from outside elements, not disrupting the worker while performing the work tasks.

*3.3. Rapid Upper Limb Assessment (RULA)*

An anthropometric working environment refers to the conditions that should be provided to a person at the workplace so that the work performed is maximally adapted to that specific person's anthropometric (dimensional) characteristics. In ergonomics, anthropometric data is used to ensure that a machine or environment is adapted to a person's physical characteristics. The main goal of risk factor assessment is to increase comfort and reduce pain and the occurrence of musculoskeletal disorders. RULA [16], addresses this issue with its score calculator for upper body risk assessment. It is designed to assess the force, posture, and movement associated with sedentary tasks, including manufacturing, retail, computer tasks, laboratory work, or where the individual is seated or standing without moving or performing static or repetitive work, which may contribute to muscle fatigue. An experienced ergonomist can rapidly obtain an evaluation for individual workers and use this data as a recommendation pointer to workplace redesign projects and productivity estimates. An interview and observation of the work routine cycles are beneficial to understand better working place conditions. The drawback of such assessments is that they require the evaluator's experience. Because the worker knows that he or she is being observed and may not behave or perform actions as he or she usually does in practice, he or she is cognitively more burdened and exposed to more significant stress, so there is a possibility of compromising the recorded data. In this sense, we see the need to introduce discrete types of monitoring in the form of wearable technology that will support this kind of risk assessment methodology.

RULA assessment tool requires an ergonomics evaluator to determine the postural angles of six different body positions. In most cases, the experienced evaluator can determine the body position angle in the field as he or she observes the work task. However, it is recommended by the creators of the tool to take pictures or videos of the work task being performed from several angles if possible. The evaluator can display the pictures on the computer monitor, either print them down on paper and use a goniometer or an overlaid transparent protractor image to measure the body segment angles. This evaluation approach can be classified using optical sensors to get representative risk assessment data. A wearable upper body flex sensor is suitable to perform the same evaluation task providing the evaluator, and the wearer with in-time information on trash holds risk limits exceeds. RULA is based on the score chart methodology with an action level output that identifies an indication of urgency, postures are studied, frequency of muscle use and force exertions are recorded, and each of these elements gets a corresponsive score value resulting in the final score that represents a relative score of ergonomics risk not an absolute score of risk, for absolute ergonomics risk estimate evaluator should additionally consider all risk factor elements of a complete workplace environment. A good trait of this methodology we have exploited here is its quick implementation in the evaluation practice and rapid data extraction, which provides ergonomics evaluator with the possibility to compare pre- and post-results of particular interventions and redesigns in the work procedures. Although it is very important in the ergonomic design of the workplace, the RULA methodology

that we are considering here does not cover parameters of psychological and physiological stress levels.

Neck and trunk posture are usually interconnected in a work environment when it comes to ergonomics risks and should be parallelly observed when it comes to interventions, scoring assessments for these two postures are done separately, and the sum of their combined scores is later used in RULA calculator. In our experiment, we score the neck position with 1 to 4 points, Figure 1. The score is based on the degree of neck flexion or extension. Neck flexion is a movement of the chin towards the chest from a neutral neck position. Neck extension moves the chin away from the chest, backward from a neutral neck position. Experts in biomechanics use a variety of landmarks and methods to define the zero point between flexion and extension or neutral position of the neck.

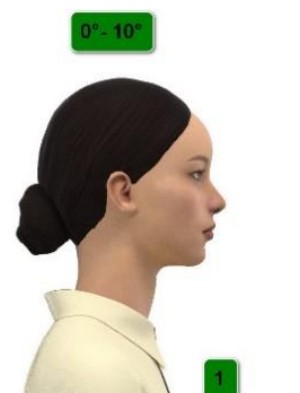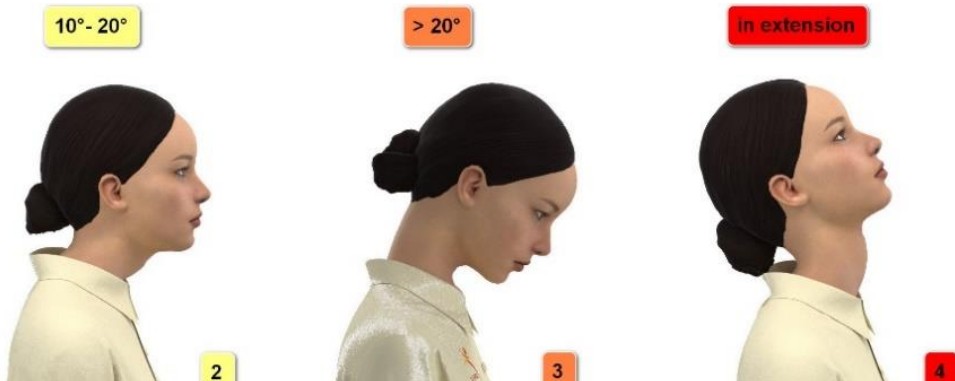

**Figure 1.** Neck positions with the score points for the angle of flexion or extension.

After the flexing degree of the neck is determent, the next factor to observe is muscle use which needs to be identified, and if it is present, it brings 1 point, and if not 0 points are added to the equation. Muscle use is determent by the following condition if the observed posture during the task is mainly static, held for more than 1 min, or if the action repeated occurs four times per minute, it scores 1 point. If neither condition exists, no entry is made for the muscle use score.

The trunk position score points will be between 1 to 4, Figure 2. The score is based on the degree of trunk flexion or extension. Trunk flexion is defined as the anterior, forward movement of the trunk in the sagittal plane, while trunk extension is defined as the posterior, backward movement of the trunk in the sagittal plane, which is outside of our assessed scope.

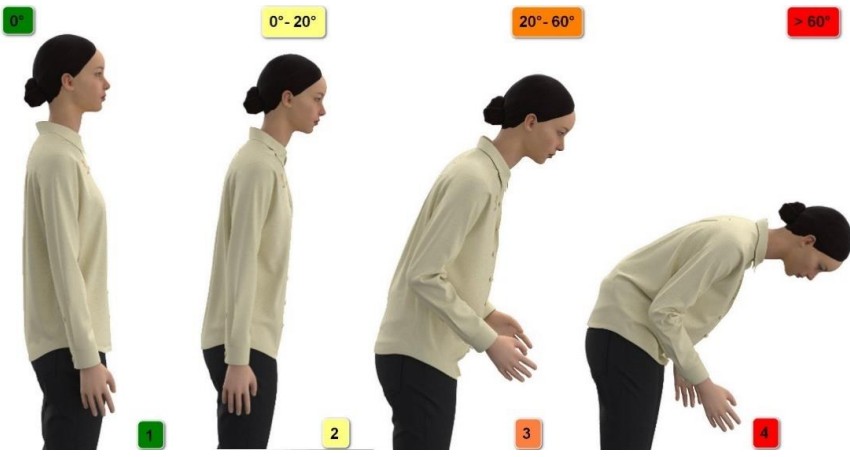

**Figure 2.** Trunk positions with the score points for the angle of flexion.

Fore is the last score is the last parameter that we calculate in this ergonomics risk assessment method. Force represents the force or load values required to finish the task by the person who is observed. Force is declared in weight units' kilograms or pounds, and we use the dynamometer to measure the force implied to start, maintain and finish the task. It is essential to separate measurements between the different frequencies of forces introduced during the task process, and we need to identify three types of forces that are occurring in this form for scoring: intermitted, repetitive, and static force. Intermitted occurs irregularly < 4x repetitions per minute and does not carry a load over 2 kg it scores 0 points, intermitted occurs irregularly < 4x repetitions per minute with a load of 2 kg to 10 kg this one scores 1 point. Static or repeated when held > 10 min, or if action is repeated occurs 4x per minute with a load of 2 kg to 10 kg scores 2 points, if more than 10 kg is measured or if repeated or shocks such as hammer tool usage, this occurrence scores 3 points.

The first evaluation score to determent is posture state Figures 1 and 2, both for trunk and the neck posture during the task, the sum of two values is entered into calculation after this is done next step is to evaluate if there a condition for muscle score in Table 1, if it is present for one or both postures the sum of the scores is entered for calculation, and the last parameter is the force/load score following the quite from Force score table Table 1, we should be able to determent the adequate score point and enter it for calculation in the same manner like described above for previous parameters. The RULA calculator has dedicated fields to enter or choose values for each score for the final calculation and assessment of the ergonomics risk.

POSTURE SCORE + MUSCLE USE + FORCE = ACTION LEVEL SCORE

**Table 1.** Muscle use and force scores reference guide.

| Score | Muscle Use Scores Table |
|---|---|
| 0 | No condition present |
| 1 | Postures that are mainly static (held for longer than one minute) Repetitive use (actions repeated more than 4 times per minute) |
| **Score** | **Force scores table** |
| 0 | weights or forces $\leq$ 2 kg and held intermittently |
| 1 | weights or forces 2 to 10 kg) and held intermittently |
| 2 | weights or forces 2 to 10 kg and held statical weights or forces 2 to 10 kg and repetitive weight or forces $\geq$ 10 kg and held intermittently |
| 3 | weights or forces $\geq$ 10 kg and held statically weights or forces $\geq$ 10 kg and repetitive shock or force with rapid buildup such as hammer use |

After getting the final score, we turn to the action level chart Table 2, to follow its guidelines and check if the action level suggests any modification. If it does, they should be made and retested for validation until we are at the safe level. For our research, we are using these scores to compare results received when the worker is wearing the sensor with the situation when he is not using the sensor under the same working environment and task procedures. If we can stay at the safe levels, that would prove that the sensor can be an alternative to often costly workspace and procedures redesign.

**Table 2.** RULA action level ergonomics risk assessments and guideline.

| | |
|---|---|
| **Action Level 1** | Score of 1–2 = **Acceptable**, negligible risk, no action required |
| **Action Level 2** | Score of 3–4 = **Low risk**, Investigate further |
| **Action Level 3** | Score of 5–6 = **Medium risk**, investigate further and change soon |
| **Action Level 4** | Score of 7 = **Very high risk**, investigate further and change immediately |

## 4. Results

Before starting the experiment, the team familiarized itself with the basic actions that take place at the observed assembly line workstation. All technical operations performed by the worker during the entire process are categorized. An assessment was made of secondary ergonomic factors related to the design of the workstation and environment so that they would not affect the outcome of our research. These factors relate to the height of the work surface from the floor on which the worker is standing, the distance of the worker's body from the edge of the work surface, the stability of the surface on which he is standing, the level of noise, light, ventilation, humidity, and temperature. The position of the arms and the angle of the joints of the elbows, hands, fingers, shoulders, hips, and legs were beyond the scope of this research, but they were also considered, and adequate corrections were made so that these factors did not affect the outcome of the research. All these factors individually are significant for assessing the overall ergonomic risk, each requiring a separate assessment. The work task that we monitored is of a static type and takes place on a press in which the worker places assembly product pieces, the time cycle of the worker's operations is dependent on the machine time of the press required to operate, which is 5 s, product parts handled by worker weigh up to 200 g, and every recorded power loads wore under 2 kg threshold mark, the press is of a continuous type, and the worker does not have direct physical interaction with it. The height of the work surface from the floor is 110 cm, and the worker's body is 10 cm away from the edge of the work surface. He performs production operations while standing, using both hands simultaneously.

### 4.1. Assessment of Ergonomic Risks in Current State Workplace

We started the research with the conventional assessment method according to the instructions of the RULA tool. The worker was monitored visually while photographs and videos were taken at different periods of the shift for later analysis by the team, and based on them the recorded ergonomic risk condition was scored. Scoring is presented by the marked hour, representing the average score value before that mark hour. As expected, the scoring results fluctuated throughout the time laps of the work shift, the highest peaks of oscillation occurred before the first break and before the end of the shift, with a general tendency of growth over time. The average score we determined for this workplace and worker is 5. As such, it has scored in the medium risk zone of ergonomic injuries according to RULA action level guidelines Table 2. It requires intervention regarding the reorganization of the work process and procedures.

The average points of the assessment of the workers by the elements of the examination are as follows:

- Neck posture score: 2;
- Trunk posture score: 2;
- Muscle use score: 1;
- Force/Load score: 0;
- Final score: 5; (investigation and changes are required soon)

The production process is performed in a repetitive style more than four times per minute and receives a score of 1 for muscle use, while all recorded forces were below 2 kg, so force load scores 0 points in the calculation. Scores for neck and trunk pose were established as the most critical, with an average score of 2, but in certain moments they reached a score of 3. It was noticed that after a certain time, the worker has difficulties maintaining the correct body position due to fatigue and loss of concentration, this manifests in lowering the chin forward and down towards the assembly product, which causes the whole body to move forward and leads to a change in the angle of holding the trunk in an ergonomically unfavorable situation.

After analyzing the obtained data Table 3, it was determined that the possibility of redesigning the workstation must be considered, as well as the work procedures themselves, as much as possible in practice, considering that it does not affect the plant's productivity. The most economical possibility to reduce the established risks is additional training for

the worker himself, the challenges of this approach are the time required for training as well as constant supervision after the training because physiological variables cannot be avoided, and the worker will always be prone to fatigue and lack of concentration to monitor the ergonomic parameters risks independently. The next consideration is the rotation of workers at this assembly line workstation during the shift. This approach requires workers to be trained for more manufacturing operations, if possible and suitable, this method is most often used in industrial ergonomics practice. Its challenges are worker training, productivity, and quality control of the production itself.

**Table 3.** Average score points assessed in different time intervals during the shift.

|  | 7 am | 9 am | 11 am | 13 am | 15 am |
|---|---|---|---|---|---|
| **Posture score** | 3 | 4 | 4 | 4 | 5 |
| **Muscle use** | 1 | 1 | 1 | 1 | 1 |
| **Force/load** | 0 | 0 | 0 | 0 | 0 |
| **Risk score** | 4 | 5 | 5 | 5 | 6 |

The last consideration is changing production procedures, adapting production equipment, or purchasing more ergonomically adequate tools and equipment. This approach, if possible, is perhaps the most effective in the long run. However, it is the least economical because it requires additional investments in equipment and services, time to accept and implement new production procedures, often causes a temporary suspension of production, possible changes in the management and logistics system, and as such, is generally the least popular in the decision-making process.

This analysis showed us the challenges of the specific assembly line workstation and which challenges should be addressed most efficiently. We found that the worker cannot independently monitor all conditions of ergonomic risks and requires constant external control to stay out of the risk zone Figure 3. The work is static and subjected to loss of concentration. We also established that the ergonomic risks of the workstation itself increase over time, which is in direct correlation with the increase in fatigue of the worker. In the following, we will respond to these challenges by introducing a sensor for monitoring the body's position, which will notify the person wearing it with a signal when certain value thresholds are exceeded and when he or she is in a risk zone susceptible to ergonomic injuries.

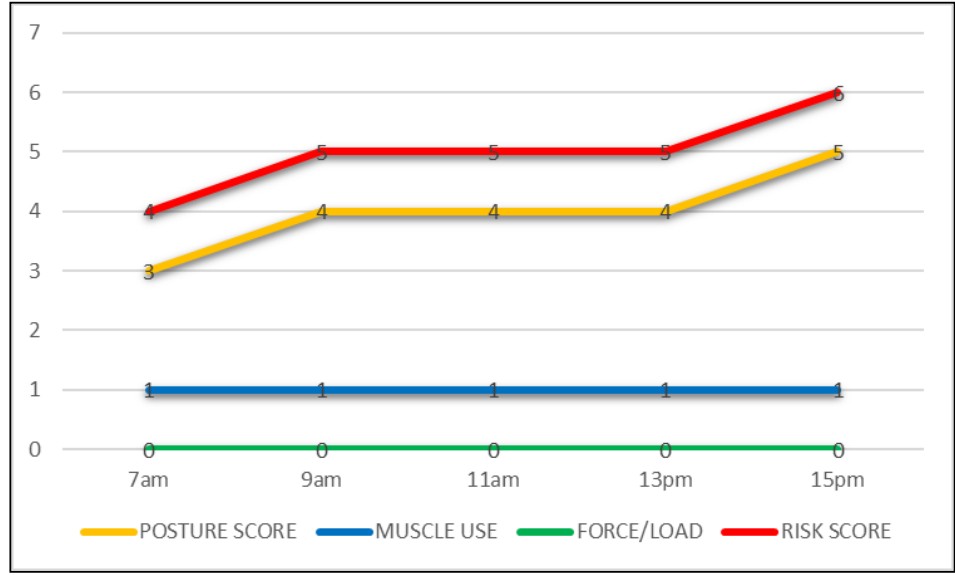

**Figure 3.** The graph shows the progression of ergonomics risk true time.

### 4.2. Assessment of Ergonomic Risks with the Person Wearing the Motion Sensor

To answer the challenges of the previous research, we will use a workwear garment in which a motion-tracking sensor is embedded. The sensor is installed on the back piece of the workwear t-shirt Figure 4 and works on the stretch detection principle. Stretching occurs in situations where the wearer bends his back, the garment on which the sensor is located is stretchable enough to have the feature of following the curvature of the backbend and therefore increases the length of the rear piece of the garment. This new condition is detected by the microprocessor that reads the condition at the sensor, the received signal is processed, and based on it, a signal is sent in the form of sound and LED lights. The sensor, microprocessor, and battery are embedded together in the clothing in such a way that they allow free movements and do not hinder the worker when performing the tasks.

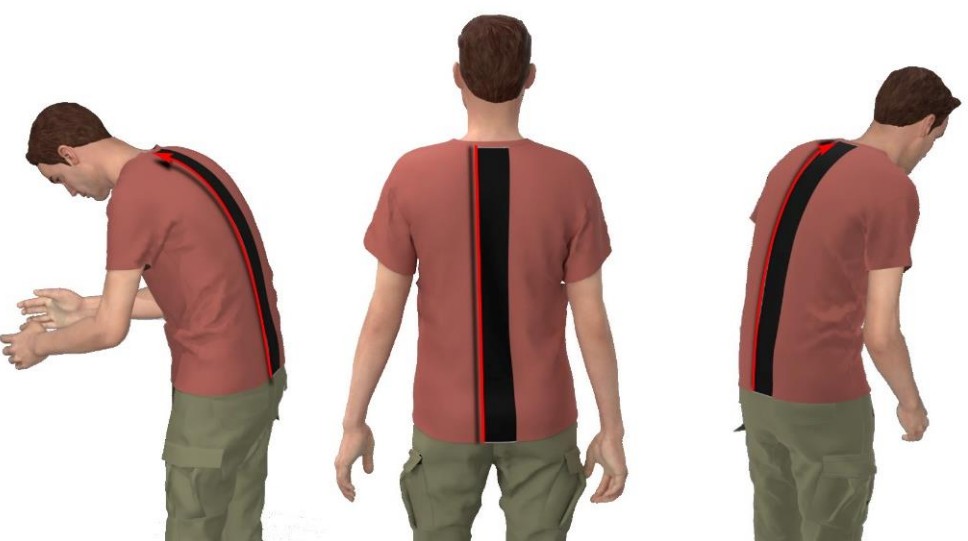

**Figure 4.** Placement and function of the motion sensor.

The assessment was performed the next day after the previous analysis, and the same worker was wearing the garment with the sensor when he started his shift. We performed the assessments as before using the RULA guidelines. The observations were recorded with photos and videos for later analysis in a predefined manner.

This time the worker was signaled by the wearable every time his current pose was outside the acceptable range of ergonomic risks. He would then simultaneously and independently correct his body position. The worker was previously familiarized with functions and had the opportunity to try the wearable sensor, so he did not have a period to get used to it. Although the requirements of the work task itself were such that the worker was put in situations where he produced an unfavorable body position, he consciously corrected his posture in most cases. In situations where the work operation would require the worker to lean his body forward, which is generally accompanied by the person lowering his chin down towards the object of production, which causes an even greater level of curvature of the neck and back, after the sensor signal, the worker would raise his chin and straighten his neck which was followed by the sound signal stopped.

After the data was collected and during the analysis, it was noticed that the changes occurred in the posture score rank, whose average for the entire shift was now 2.4 score points. In comparison, the total ergonomic risk average was 3.4 score points, and it is categorized as low-risk activity. This result raised the level of worker protection from ergonomic risks by one ladder higher according to RULA standard action level guidelines Table 2.

The average points of the RULA assessment of the workers by the elements of the examination are as follows:

- Neck posture score: 1;
- Trunk posture score: 1;
- Muscle use score: 1;
- Force/Load score: 0;
- Final score: 3; (further investigation is needed, and changes may be required)

Although we have not yet achieved a result that guarantees the targeted highest level of ergonomic protection, we managed in a short period to obviously reduce the level of ergonomic risks with no changes to the processes that take place at the workstation and without directly disturbing the assembly line procedures of the worker himself. After processing the obtained data, the next step is consideration and improvement proposals for further adaptation of the workstation to ergonomic requirements, after which a new assessment is performed according to the previous procedure with an aim to re-evaluate and confirm any benefits obtained in terms of worker protection. Re-evaluation of ergonomic risks should be performed every time after the change is made at the workstation or procedures.

The wearable helped the worker correct his body posture independently. At the same time, it made the work of the ergonomic assessors easier, even though they used RULA assessment methods for the final result Table 4, it helped in the overall assessment of the observed procedure and allowed them to notice certain anomalies that might have been previously neglected. As explained earlier, the sensor is calibrated according to the neutral position of the body, any deviation is alarmed by a signal so that improper ergonomic actions of the worker are now observed while he is not performing the work task and is not at the observed workstation, and these additional actions also affect in the buildup of ergonomic risk score, however, they mostly remain unrecorded during conventional monitoring.

**Table 4.** Average score points are assessed when worker is wearing the sensor.

|                | 7 am | 9 am | 11 am | 13 am | 15 am |
|----------------|------|------|-------|-------|-------|
| **Posture score** | 2 | 2 | 2 | 3 | 3 |
| **Muscle use** | 1 | 1 | 1 | 1 | 1 |
| **Force/load** | 0 | 0 | 0 | 0 | 0 |
| **Risk score** | 3 | 3 | 3 | 4 | 4 |

As we can see when analyzing the graph Figure 5, the fatigue factor of the worker remains present throughout the period, the worker received signals from the sensors, but his reactions to them were slower over time, and as a result, he stayed longer in the zones of ergonomic risks the longer he spent at the workstation, it is evident that the most critical time is towards the end of the working day when the worker is already tired and becoming mentally less concentrated on work tasks. This information should be taken seriously when considering dangerous workplaces with a tremendous potential for physical injury to workers, which is certainly a more significant risk factor than ergonomic injuries. Ergonomics-related conditions are characterized by developing over time spent under established unfavorable norms. When analyzing ergonomic risks, if a situation is established as such that the work is performed within non-ideal but tolerant limits, as is the case in our research, broader aspects of occupational safety conditions should be considered.

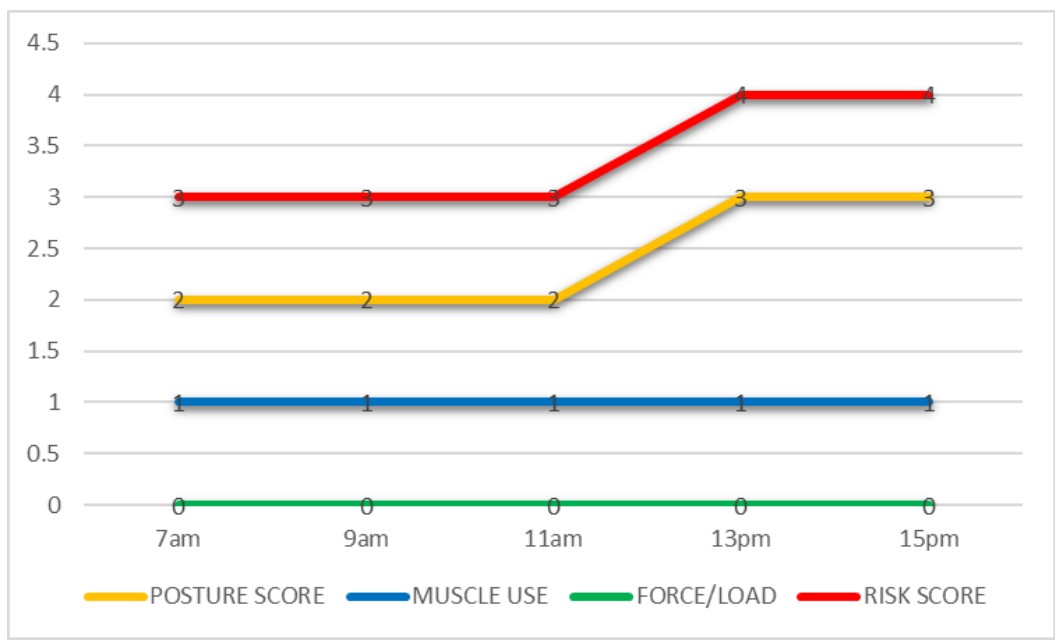

**Figure 5.** Graph shows improvements in reducing the ergonomics risks.

As can be seen in Figure 6. the worker performs the task at the observed workstation. In the moments when the posture of his body is in the zone of acceptable risk, the LED located at the top of his back is turned off, and that position in the picture is marked as "Good." At the moment when his posture moves into the zone of increased ergonomic risk, the LED lights up red, and the sound signal is activated; this is marked as "Bad" in the picture.

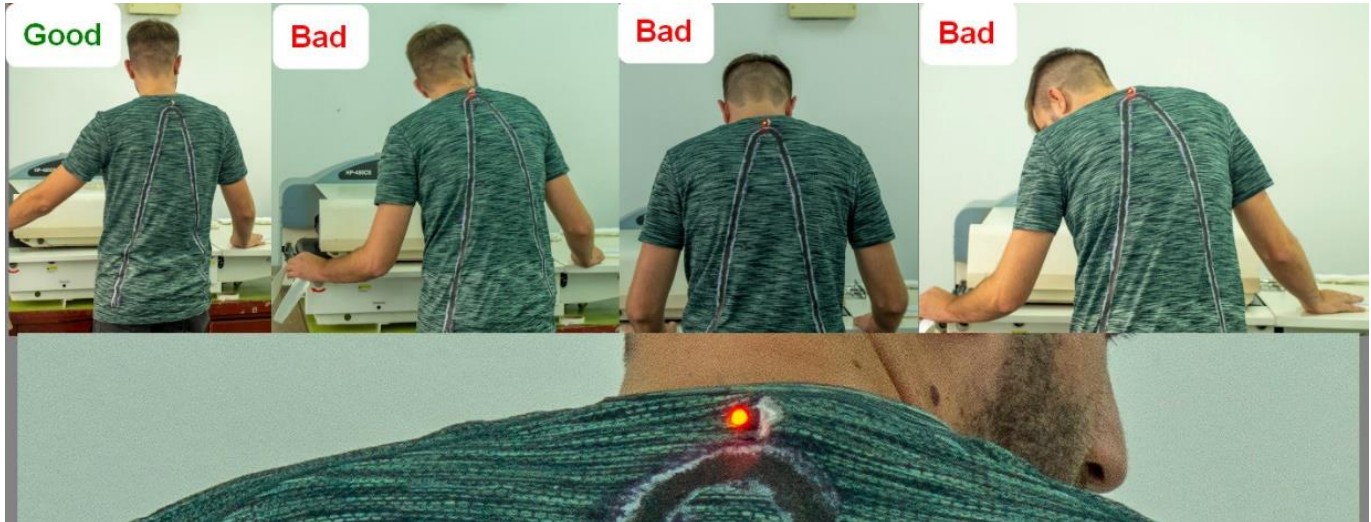

**Figure 6.** Assessed activity and wearable's LED signaling when the person is in a risk-prone posture.

## 5. Data Analysis

The analysis of recorded data was performed by comparing two sets of results obtained in separate scenarios, the first when the worker does not use the wearable sensor data collected in group 1, and the second when he uses it group 2. Data recording was done on different days for each scenario at the workstation presented in this research. The most influential parameter for this experiment was the posture score according to the RULA methodology. The ergonomic risks of body posture parameters are evaluated according to

the method described in Section 3.2. regarding the risk thresholds described in Figures 1 and 2. The analysis covers the overall ergonomic risks score for both cases and their comparison. Observation score results are shown in Tables 3 and 4.

To analyze these two sets of data, we chose the two-tailed t-Test statistical method to estimate the difference between two data set group means using the ratio of the difference in the set group means over the pooled standard error of both set groups. We use here a two-tailed t-Test because it is used to measure the difference between precisely two means which is suitable for evaluating the results of our research and the hypothesis that we have set.

During the recording of the work task, when the worker performs normally and does not use the wearable sensor, a higher level of ergonomic risks was established based on the RULA score chart Table 2, this was confirmed by analyzing the recorded data for the entire 7.5 h shift (mean = 5, standard deviation = 0.6325). Mean 5 ranks this working position a medium ergonomic risk Table 2, the standard deviation is affected by the worker's physical performance and mental focus during the time.

The next day, the same worker was observed performing the same work task under the same conditions. This time different results were recorded after assessing ergonomic risks (mean = 3.4, standard deviation = 0.4899). A mean of 3.4 ranks this result as low ergonomic risk, while the standard deviation indicates improvements in the worker's mental focus regarding body posture.

To find t calculated difference value and degrees of freedom, we will use the following formulas:

$$\overline{X}_1 \approx 5,$$

$$\overline{X}_2 \approx 3.4,$$

$$S^2X_1 = \frac{1}{n-1} = \sum_{i=1}^{n}\left(X_{1i} + \overline{X}_1\right)^2 \approx 0.4,$$

$$S^2X_2 = \frac{1}{n-1} = \sum_{i=1}^{n}\left(X_{2i} + \overline{X}_2\right)^2 \approx 0.24,$$

$$S_{X1X2} = \frac{1}{n-1} = \sqrt{\frac{1}{2}\left(S_{x1}^2 + S_{x2}^2\right)} \approx 0.5657,$$

$S_{x1}$ = Standard deviation of data for group 1, $S_{x2}$ = Standard deviation of data for group 2, $S_{x1}x_2$ = Grand standard deviation.

$$\frac{\overline{X}_1 - \overline{X}_2}{S_{X1X2} * \sqrt{\frac{2}{n}}} = \frac{5 - 3.4}{0.5657 * \sqrt{\frac{2}{5}}} \approx 4.472,$$

$$\text{d.o.f} = 2\,n - 2 = 2 * 5 - 2 = 8,$$

The means of group 1 and group 2 are significantly different at ($p < 0.05$). Calculated difference t for these data sets is (t = 4.472) to determine the critical value for t we use degrees of freedom (d.o.f = 8) and ($\alpha = 0.05$). The critical t value we get here is 2.306, less than t = 4.472. This tells us that the means are significantly different.

This result suggests that there is a noticeable difference between these two scenarios Table 5. When we analyze means and standard deviations, it is evident that the worker's posture shows improvements in reducing ergonomic risks during the time spent at the workplace. This is caused by his increased focus on body posture during the performance of work tasks when wearing the sensor.

There is a distinction in the trend when comparing the two scenarios, differences in ergonomic risk scores at the beginning and the end of working hours are smaller when the wearable sensor is used.



**Table 5.** Results of the data analysis, group 2 represent data when using the wearable.

|  | Group 1 | Group 2 |
|---|---|---|
| Mean | 5 | 3.4 |
| Variance | 0.4 | 0.24 |
| Standard Deviation | 0.6325 | 0.4899 |
| n | 5 | 5 |
| t | 4.472 | |
| d.o.f | 8 | |
| critical value | 2.306 | |
| **t > criticall value—there is significant difference** | | |

## 6. Discussion

The research results indicate that wearable users tend to show improvements within acceptable risk limits. At the same time, ergonomics safety supervisors are provided with better quality and quantity of real-time data for their analysis. The results show that introducing a wearable sensor that monitors body movements as a preventive measure to reduce ergonomic risks has its benefits in reducing and detecting ergonomic risks in the work environment. Using the established RULA method and comparing the results obtained in the same way, but in different circumstances, this time using the wearable sensor, we managed to prove in practice that there are differences in the final evaluation results when wearable technology is used to monitor and correct risk factors determent by established methods. This kind of technology opens numerous other upgrade possibilities, and our proposal is an open architecture kind providing the possibility of installing sensors in series or other types of sensors as well as communication modules that can turn the wearable into a kind of IoT device and thus open a wide range of inspiring possibilities. Different types of sensors can be used for wearables with the purpose of preventing ergonomic risks, they can be divided into two main groups: a collection of physical and physiological data. When collecting physical parameters, motion sensors, stretching or optical sensors are generally used. They mainly consist of a measuring sensor and a processing unit that monitors the deviations of the collected data. Heart rate sensors, sensors for chemical analysis of blood, sweat and oxygen consumption can be used for electronical collection of physiological parameters. Physiological parameters can also be monitored using smart e-textile fabrics whose chemical and physical properties can be used for analog data analysis. The very structure and materials used for weaving or knitting the e-textile fabric may be physically or chemically reactive to deviations in the environment in which it is used. Of course, this technology also brings a certain number of challenges. From the point of view of the garment design, there are challenges in the installation and maintenance of the item itself. These challenges relate to the choice of appropriate materials and the technological process of clothing production itself. Specific procedures in this technological process must be adapted for the installation of electronic components in the garment and often imply the adaptation of traditional production equipment and procedures to new requirements. One disturbance was observed that could affect the cognitive state of the worker, and it was the sound of the warning buzzer that warns of unwanted body positions. After some time, it was noticed that it occasionally attracts the worker's attention. For our experiment, as previously described, a sensor used that uses electrical resistance, a property of the material for its reading and value conversion. The challenges we encountered were reflected in matching the properties of electronics, which mainly exist on rigid and stable surfaces, with the physically flexible properties of textile materials. This limits the matching, as well as the service life of the installed components. The next challenge that arises comes in the form of the physical characteristics of the material and their connection with the resistance itself, as it is known that the material's electrical resistance varies not only depending on the type of

material but also on external factors such as temperature and humidity. The human body, by its nature, generates temperature and humidity in the form of sweating. The work environment can also contribute to these factors. Another challenge is the physical sensitivity of electronic components and their tendency to break and malfunction. The worker in his work environment often comes into contact with various objects that surround him, and as a result, the components on the wearable itself may fail. There are still several challenges that must be approached individually and methodically in future research.

## 7. Conclusions

This research aimed to assess the electrical resistor's usability as a valid sensor of human movement when embedded in regular clothing items. The results we obtained confirm its prospective usability. Research has proven this by evaluating results obtained using a sensor and comparing them with the results obtained under the same conditions but in a conventional way. One goal of our research was to prove that there is a justified cause and possibility for further development and employment of wearable technology for the needs of occupational safety, and we believe that we have proven this claim through this narrowly focused experiment, through whose work and results from we have received inspiration for further research in that direction. Modern industry is increasingly demanding effective ergonomic solutions for its workers. It stems from humane as well as productive motives. It is important to note that ergonomic conditions have two branches, physical and cognitive. It is a well-known fact that many workers are absent from work or work with difficulties due to illnesses or conditions caused by work activities. Ergonomic disorders are the fastest-growing of all categories of occupational diseases. 30% to 50% of workplace injuries are related to ergonomics in some way. Some of the most affected are employees in the manufacturing, retail, and service industries. Of most significant importance is the prevention and reduction of ergonomic risk related to physical stress factors and workplace conditions that carry the risk of damage or disease to the musculoskeletal systems of employees.

**Author Contributions:** Conceptualization; methodology; validation; investigation; resources; data acquisition; writing—original draft preparation, funding acquisition, N.M.; writing—review and editing; supervision; project administration, M.Č. and M.T.; formal analysis, D.R. All authors have read and agreed to the published version of the manuscript.

**Funding:** The authors would like to thank RAMAX Company DOO, Pozega, Serbia for their outstanding support enabling this research possible using their production facilities as well as for their financial support.

**Data Availability Statement:** Not applicable.

**Conflicts of Interest:** The authors declare no conflict of interest.

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
