# Peer review of "Challenging Ergonomics Risks with Smart Wearable Extension Sensors"

_electronics, doi:10.3390/electronics11203395_

Round 1

Reviewer 1 Report

It is requested a major review:

- errors in references. For instance RULA is indexed as [15] when it is [16]

-Basically methodology is well organized but there are some aspects that are not powered from the research view. In particular, the ones regarding to the technology. Some major aspects are:

Section 3.1 gives a vague idea of the technology adopted including when it is referred to CEN standards (what is the set of them? Any references in the work)

Section 3, 4 and conclusions: there are not evidences of the use of technology sensors to the conclusions more than an assessment using LED in section 3.1. In particular Figure 6 seems to be referred to a wearable LED, where is it in the picture to describe the action as good or bad?

Author Response

Response to the comments - Reviewer 1

Reviewer 2 Report

Краткий обзор статьи
«Борьба с рисками эргономики с помощью умного носимого удлинителя»

В представленных к публикации материалах рассмотрены некоторые выделенные авторами особенности использования технологий Интернета вещей в интересах охраны труда.

С одной стороны, одной из сильных сторон выбранной и предложенной темы исследования является отчасти достаточная «погруженность» изыскателей в основные вопросы данного исследования, а также важность темы и значимость данного направления в практика охраны труда в целом.

However, on the other hand, the insufficiently clear statement of the purpose of the article, the scientific problem being solved (justification and argumentation of limitations and assumptions as in a new problem), its ambiguous structuring and argumentation, as well as the style of presentation do not allow us to fully appreciate all the expected advantages of such an interesting development.

Comments are not focused on presentation, argumentation, or English language deficiencies, as they should be addressed at a later stage with the help of editorial staff. But to eliminate weaknesses, it is desirable to ensure greater consistency of the text of the article.

The main comments on the general concept – in the interests of potentially reducing the "the ergonomics risks" – can be partially explained, including by ensuring the completeness of the covered review topic for the reasonable use of an "smart wearable extension sensor". Here it is necessary to pay attention to a more explicit emphasis on identifying a gap in knowledge, establishing cause-and-effect relationships, as well as supplementing the list of references, etc.

Taking into account the possible correction of the title of the article, it is necessary to indicate how the scientific novelty of the study of the possibilities of using this sensor will reduce the risks. It is desirable to disclose more fully in the content of the article the results of reducing "ergonomic risks", an unambiguous definition of such in relation to these conditions.

It is important to pay attention to the formulation and verifiability of the research hypothesis as a whole, which, obviously, will allow during the discussion to identify the desired methodological inaccuracies, etc. What, according to the authors, stands out in this question as original? How do the results obtained ensure progress in modern knowledge [about occupational Safety and Health, new sensors]?

So, it is important to formulate more clearly a new scientific problem to be solved in relation to the key hypothesis of the study (initial data, desired, limitations, assumptions, etc.).

Partially anticipating the consideration of the contents of the manuscript, it is important to take into account the following when discussing: 

Is this approach scientifically sound and is the experimental plan suitable for testing the hypothesis? Are the data and analysis presented correctly? Is the data reliable enough to draw conclusions? Are the results interpreted correctly? Are they significant? Are hypotheses and assumptions carefully identified together as such?

Also, please provide more detailed information about the example used in the work. Are the conclusions for this example consistent with the evidence and arguments presented?

Particular attention follows from the question: are the methods and tools described in sufficient detail to allow other researchers to reproduce the results? Is it possible in principle to reproduce the results of the manuscript based on the methods and techniques given in this article? 

Are all the conclusions justified and confirmed by the results? Please consider indicating the availability of the necessary data to ensure that they are adequate in due measure. But is it easy for an unprepared reader to interpret and understand them?

What can attract a wider readership, or will this article be of interest only to a limited circle of specialists? (See the objectives and scope of the journal "Electronics"). It is proposed to include a section describing the characteristic features of the "smart wearable extension sensor" in the context of modern advances in the field of conductive and/or intelligent textiles. So, for example, this [https://www.sciencedirect.com/topics/engineering/conductive-textile], or others. And, undoubtedly, the expected interest for the readers of the journal will be greater if the conclusions of the article take into account the interests of the prospects for further improvement of the developed sensor.

И все же некоторая избыточность в перечислении общепринятых методических рекомендаций по улучшению качества представления новых результатов исследований лишь повышает интерес к дальнейшему развитию всех полезных идей столь важного направления деятельности авторов.

Таким образом, просим выделить окончательный вклад автора в науку в совокупности результатов новых исследований датчика, используемого в интересах охраны труда.

Author Response

Response to the comments - Reviewer 2

Reviewer 3 Report

Flexible and wearable electronics is a hot research topic over the past years. The authors gave a potential application scenario for the wearable device. Based on ergonomics, the garment embedded body posture tracking sensor could be used as standard working equipment to help correct improper and high-risk upper body positions during prolonged and static work activities. It is good starting pinot,  This reviewer recommend the author to do further studies before publish, the research could be either to design the suitable wearable device and apply it on this case study, or to do real experiments to collect enough data for analysis worker behavior from psychological, physiological, and anatomical respects. Give the guideline for the prevention and reduction of ergonomic risk related to physical stress factors and workplace conditions

Author Response

Response to the comments - Reviewer 3

Reviewer 4 Report

The paper deals with a very interesting matter, but in my opionion it suffers from a low Scientific Soundness.

The Authors are encouraged to review their work taking into account the following suggestions, in order to improve it for publication:

1) No scientific derivations are reported and accurately described in the text, to justify the proposed procedure of investigation;

2) In order to optimally characterize a generic used sensor (here "smart wearable extension sensor"), it is necessary to indicate its relevant metrological characteristics and/or parameters, suitably tested in repeated experiments;

3) All the results have to be carried out downstream of an adequate statistical investigation, reporting the statistical parameters of interest (distribution, number of investigated points, mean, standard deviation, etc.), in order to make readers understand the significance and reliability of the results achieved;

4) The limit of the proposed procedure have to be clearly indicated, eventually as an analytical function of specific boundary conditions.

In conclusion, the paper have to be rewritten as a major revision, to make it a good scientific work, suitable for publication.

Author Response

Response to the comments - Reviewer 4

Round 2

Reviewer 1 Report

Section 3.1. It has not been solved the references to European Standards, their identification and application to the purpose of this paper. 

Figure 6 (Assessed activity and wearable’s LED signaling when person is in risk prone posture) is lost in the new version

Author Response

Response to the Reviewer 1 comments

Reviewer 3 Report

Based this revision, the authors gave more details for a potential application scenario for the wearable device. the same comments from this reviewer. this paper more suitable for healthcare journal rather than electronics, since it is is not development of electronics devices, materials etc.

Author Response

Response to the Reviewer 3 comments

Reviewer 4 Report

Even if the paper may be improved on the basis of further metrological considerations (see my previous review), I think it can be published.

Author Response

Response to the Reviewer 4 comments

Round 3

Reviewer 1 Report

All issues are properly solved

Reviewer 3 Report

same comment as brfore this paper more suitable for healthcare or application type journal rather than electronics.